# A Sensing System Based on Public Cloud to Monitor Indoor Environment of Historic Buildings

**DOI:** 10.3390/s21165266

**Published:** 2021-08-04

**Authors:** Zhongjun Ni, Yu Liu, Magnus Karlsson, Shaofang Gong

**Affiliations:** Department of Science and Technology, Campus Norrköping, Linköping University, SE-601 74 Norrköping, Sweden; yu.a.liu@liu.se (Y.L.); magnus.b.karlsson@liu.se (M.K.); shaofang.gong@liu.se (S.G.)

**Keywords:** Internet of Things, edge computing, cloud computing, historic buildings, indoor environment

## Abstract

Monitoring the indoor environment of historic buildings helps to identify potential risks, provide guidelines for improving regular maintenance, and preserve cultural artifacts. However, most of the existing monitoring systems proposed for historic buildings are not for general digitization purposes that provide data for smart services employing, e.g., artificial intelligence with machine learning. In addition, considering that preserving historic buildings is a long-term process that demands preventive maintenance, a monitoring system requires stable and scalable storage and computing resources. In this paper, a digitalization framework is proposed for smart preservation of historic buildings. A sensing system following the architecture of this framework is implemented by integrating various advanced digitalization techniques, such as Internet of Things, Edge computing, and Cloud computing. The sensing system realizes remote data collection, enables viewing real-time and historical data, and provides the capability for performing real-time analysis to achieve preventive maintenance of historic buildings in future research. Field testing results show that the implemented sensing system has a 2% end-to-end loss rate for collecting data samples and the loss rate can be decreased to 0.3%. The low loss rate indicates that the proposed sensing system has high stability and meets the requirements for long-term monitoring of historic buildings.

## 1. Introduction

The preservation of historic buildings aims to mitigate the deterioration of façades, architectural structures and housed valuable artworks as most as possible [1]. Previous studies on the preservation of museums [2,3,4,5], galleries [6], churches [7], and cathedrals [8] have indicated that improper maintenance can result in large fluctuations in temperature and relative humidity and, hence, cause irreversible changes in dimensions of artworks. Furthermore, many historic buildings, such as theaters and museums, are still open to the public for holding activities or visiting. Human comfort also requires a suitable indoor environment, which makes it more difficult for conservation [2]. Monitoring the indoor environment can help identify potential risks and provide guidelines for improving the control strategies of heating, ventilation, and air conditioning (HVAC) systems [3,4,6] to provide suitable conditions for slowing down the deterioration. Therefore, concerning preserving artworks and achieving human comfort, implementing a system for monitoring indoor environment of historic buildings is both essential and challenging.

Monitoring the indoor environment usually includes the process of data collection, transmission, storage, analysis, and application. Many previous research studies have focused on one or more of them. For data collection, Diego et al. [9] used open-source hardware and sensors to monitor heritage buildings. Lombardo et al. [10] and Grassini et al. [11] proposed monitoring solutions based on small sensing nodes. Low energy consumption is another important property in data collection, which is considered by Huynh et al. [12], Zhang et al. [13], and Perles et al. [14]. For data transmission, Burri et al. [15] proposed a data gathering protocol that enables ultra-low power consumption. Several works used wireless sensor networks in their remote monitoring systems [12,13,16,17]. For data storage, Fernandez et al. [18] presented the database for storage used for cultural heritage preventive conservation. For data analysis and application, many research studies applied collected data to multiple aspects for the preservation of historic buildings, such as predicting indoor particulate matter [19] and modelling the thermo-hygrometric conditions [20].

Previous monitoring systems are usually specific solutions for specific application scenarios. These systems rely on particular technologies and platforms but are not designed for general digitialization and smart service purposes. When data are in focus, advanced technology such as artificial intelligence with machine learning can be utilized. Thus, a general digitalization framework for monitoring indoor environment of historic buildings is needed. Several previous studies have been reported on designing such framework. Guo et al. [21] designed an integrated monitoring system based on web services to facilitate system integration between heterogeneous data sources. Bolchini et al. [22] proposed a methodology for monitoring and data analysis of smart buildings based on existing frameworks but lacked the consideration for scenarios of preserving historic buildings. Akrivopoulos et al. [23] presented a platform that provides an integrated solution for real-time monitoring and management of educational buildings. Liu et al. [24] proposed a data-centric Internet of Things (IoT) framework that is based on a unified device management model and data model with the help of public cloud services, hence, improving the interoperability and compatibility between heterogeneous hardware and communication protocols. However, frameworks proposed in these papers are either hard to be reused by other people because of the usage of private clouds [21] and self-developed data visualization modules [24] or restricted within a small scope of functions, with a limited capability for data analysis [21], and the lack of the interface for viewing real-time and historical data [23]. Therefore, a novel digitalization framework that is adaptable, i.e., easy to be implemented and reused by other people for monitoring and preserving historic buildings, is studied and presented in this paper.

In addition to being adaptable, the new framework needs to address two challenges. The first challenge is about stable and scalable storage. Monitoring historic buildings is a long-term process, during which lots of data are collected and the system requires stable and scalable storage resources. For instance, to deploy 32 nodes in a historic building, each node needs to measure five environmental parameters every 15 s and each measured data sample is stored by two bytes, then the storage resource required per day is about two megabyte (MB). If more data are required to collect for preserving historic buildings, stable and scalable storage resources are needed. Some pioneering works used offline data loggers [8,25] or private clouds [12,13,16,21,26] to store these data. However, both offline data loggers and private clouds have shortcomings when being used for long-term monitoring of historic buildings. Offline data loggers usually store data on a single storage medium and, once the storage medium fails, all data might be lost. Hence, even with a separate local backup, the data are more vulnerable to potentially catastrophic events such as theft, fire, and events that are similar. Managing private clouds requires considerable cost and accountability. Therefore, it is necessary to find a solution to provide stable and scalable storage. Recently, the maturity of public cloud services, such as Microsoft Azure [27], Amazon Web Services [28], and Google Cloud [29], fulfills the requirements for stable and scalable storage resources. In this paper, this challenge is tackled by using public cloud storage.

Another challenge is related to the capability of performing real-time analysis to realize preventive maintenance. A significant limitation of early works [8,25] is that data analysis is usually carried out after a period of collection. Although late analysis can still provide some useful information and can improve strategies, it is better to perform real-time analysis for these data to find potential risks in time and realize preventive maintenance [18,30]. Meanwhile, to preserve historic buildings, more and more applications will be developed and deployed and, thus, the computing resources must also be scalable. Public cloud services can provide on-demand computing resources for performing real-time analysis. Moreover, edge computing is an excellent supplement to cloud computing [31] in application scenarios that have sensitive data or require low response time. Moon et al. [19] proposed a framework to predict indoor particulate matter concentration on an edge device by pre-trained machine learning models. Tse et al. [32] used a low-cost sensing system based on edge computing to monitor indoor environment of cultural heritage buildings. Recently, real-time optimization methods [33] and artificial intelligence methods [34] provide the ability for better control of the system. In this paper, cloud computing and edge computing are integrated to provide the capability for real-time analysis. With the help of public cloud services and edge devices, powerful machine learning methods can be applied to perform real-time analysis and realize automatic control, as presented by Grzonka et al. [35], in future work.

This paper focuses on overcoming the aforementioned drawbacks of existing works. An adaptable digitalization framework for preserving historic buildings is presented, which tackles the challenges in stable and scalable storage, as well as the capability of real-time analysis. A sensing system following the architecture of the digitalization framework is implemented to monitor the indoor environment of historic buildings. The implemented sensing system consists of perception devices, an edge platform, and a cloud platform. The perception devices and edge platform are packaged in a locally deployed sensor box. The sensor box can monitor indoor environmental parameters such as temperature, relative humidity, carbon dioxide (CO_2_) concentration, suspended particulate matter, harmful gases, and vibration. The obtained data are periodically synchronized to the cloud via the cellular network. The cloud part takes advantage of a series of public cloud services. A web-based user interface is developed in this study for facility managers and researchers to view the indoor environment in real-time and to check the historical trend. With the proposed system in this study, data can be analyzed in real-time either on the cloud platform to provide functionalities such as anomaly detection, pushing alerts, and preventive maintenance or on an edge device without uploading them to the cloud.

The contributions of this paper are listed as follows:An adaptable digitalization framework for smart preservation of historic buildings is proposed. This framework has a flexible architecture that other people can refer to in implementing their systems.A sensing system following the architecture of the proposed digitalization framework is implemented by using public cloud services, open-source software, and hardware. This system supports data collection, transmission, storage, and visualization. This system also supports adding more functionalities in future research.The implemented sensing system is applied on three historic buildings in Sweden for field testing. The stability for long-term operation of the sensing system is evaluated. A preliminary analysis for the indoor environment is also showcased by following industry standards.

The remainder of this paper is structured as follows. In Section 2, the layered architecture of the digitalization framework and the main components and functional modules of each layer will be provided. In Section 3, the detailed implementation of the sensing system, including used hardware, software libraries, and public cloud services, are described. In Section 4, the three historic buildings chosen for conducting field testing and the evaluation methodology are presented. In Section 5, the field testing results are illustrated and analyzed. The last section concludes the paper and discusses future work.

## 2. System Design

Figure 1 depicts the architecture of the digitalization framework. The framework consists of three parts, namely Perception Devices, Edge Platform, and Cloud Platform, following a bottom-up order.

The perception devices provide necessary data from historic buildings for the whole system. There are three categories of perception devices: Collector and Sensors, Controller and Actuators, and Broker and Other Systems. The sensors are used to measure the ambient environment. The collector is used to obtain measured environmental data from sensors. The actuators refer to devices, e.g., lighting, that can be controlled. The controller is used to receive control commands issued from upper parts and control actuators by the commands. The broker acts as an intermediary between this system and other systems, such as the building management system (BMS) in a historic building.

Each edge platform can serve multiple sets of perception devices. The communication between the edge platform and perception devices can either be through a wired bus or a wireless channel. The edge platform contains four types of functional modules, namely Aggregator, Local Storage, Local Analytics, and Local Gateway. The aggregator is responsible for communicating with perception devices and has two functions. One is gathering data, e.g., sensing data, actuator status, and the information provided by other systems, and sharing these data with other modules of the edge platform. Another is forwarding control commands to controllers and brokers. The local storage is used to store running logs, configuration files, and privacy data. The local analytics aims to analyze the collected data. The local gateway is used to communicate with the cloud platform and has two functions: One is synchronizing the aggregated data to the cloud platform and another is forwarding instructions and tasks issued by the cloud platform.

The cloud platform also has four types of functional modules, namely Cloud Gateway, Cloud Storage, Cloud Analytics, and Applications. The cloud gateway provides a hub for bidirectional communication with edge devices to synchronize data and commands. The cloud storage provides stable and scalable resources for storing data persistently. The cloud analytics provides the capability for performing data analysis tasks. Based on the collected data and analysis results, users can be provided with a series of applications, such as data visualization, anomaly detection, and preventive maintenance.

Following such a framework, researchers can make full use of resources brought by cloud computing and edge computing to design, deploy, and verify various applications for the smart preservation of historic buildings.

## 3. System Implementation

In this section, a sensing system following the proposed digitalization framework is implemented. The detailed components of each part are illustrated.

### 3.1. Collector and Sensors

A collector requires rich peripheral interfaces to facilitate sensors with different output signals. Moreover, in order to aid writing and testing programs, the collector should also provide good software support. In this study, an Arduino microcontroller board (Arduino Uno Rev3 SMD, Arduino, Somerville, MA, USA) equipped with a base shield board (Grove Base Shield V2, Seeed Technology, Shenzhen, CHN) is utilized to act as the collector (see Figure 2) to obtain readings from sensors. The Arduino board is based on the ATmega328 and has 14 digital input/output (IO) pins and six analog input pins. Six of the fourteen digital IO pins can be used as pulse width modulation outputs. The base shield board has sixteen onboard Grove connectors, four for analog inputs, seven for digital IO, one for universal asynchronous receiver-transmitter (UART) communication, and four for inter-integrated circuit communication. These Grove connectors simplify the connections with sensors.

Five sensors are used to measure temperature, relative humidity, CO_2_, suspended particulate matter, poisonous gases, and vibration in historic buildings:A temperature and relative humidity (T&RH) sensor (DHT22, Seeed Technology, Shenzhen, CHN) is used to measure temperature and relative humidity. The detecting range is −40–80 °C for temperature and 5–99% for relative humidity. The accuracy reaches up to 0.5 °C and 2% RH.A CO_2_ sensor (MH-Z16, Winsen Electronics Technology, Zhengzhou, CHN) is adopted to measure CO_2_ concentration. This sensor uses non-dispersive infrared to detect CO_2_ in the air. The measurement range is 0–2000 parts per million (PPM). The resolution is one PPM, while the accuracy is 200 PPM. The CO_2_ sensor can be operated at temperature 0–50 °C and humidity 0–90% RH.A dust sensor (PPD42NS, Shinyei Corporation, New York, NY, USA) is utilized to measure suspended particulate matter concentration in the air. This sensor can detect particles with a diameter larger than one µm. The particulate matter level in the air is measured by counting the low pulse occupancy (LPO) time in a given time unit. LPO time is proportional to the particulate matter concentration. The detecting range is 0–28,000 pieces per liter (pcs/L).An air quality (AQ) sensor (MIKROE-1630, MikroElektronika, Beograd, SRB), which carries an MQ-135 sensor, is used to detect poisonous gases, e.g., ammonia, nitrogen oxides, and benzene.A vibration sensor (Grove-Piezo Vibration Sensor, Seeed Technology, Shenzhen, CHN) based on PZT film sensor LDT0-028 is used to measure vibration and impact generated by human activities.

The program running on the Arduino board is written, tested, and uploaded with Arduino Integrated Development Environment (version 1.8.13) [36]. The native serial communication hardware UART of the Arduino board is reserved for debugging and uploading code. Therefore, the SoftwareSerial Library [37] is used to replicate the serial communication functionality. A watchdog timer is enabled to ensure the long-term stable operation of the program.

### 3.2. Edge Platform

In order to cope with the deployment conditions of historic buildings, current and subsequent research demands, the edge platform needs to meet the following requirements:Compact size and flexible mounting options for easy deployment in historic buildings;Sufficient computing and storage resources for processing and analyzing collected data or performing partitioned tasks assigned to the edge platform;Rich peripheral interfaces for connecting with perception devices.

In this study, three components (see Figure 3) are integrated to implement an edge platform to fulfill the requirements as mentioned above.

A Raspberry Pi Compute Module 3+ Development Kit (RPi CM3+ Dev Kit) (Raspberry Pi Foundation, Cambridge, GBR) is adopted as the core edge device of the edge platform. The development kit contains a Raspberry Pi Computing Module 3+ (RPi CM3+) and a Raspberry Pi Compute Module IO Board V3. The RPi CM3+ is based on Broadcom BCM2837B0, which is a Cortex-A53 (ARMv8) 64-bit system-on-chip that runs at 1.2 GHz. The RPi CM3+ has one gigabyte (GB) synchronous dynamic random-access memory (SDRAM) and up to 32 GB embedded multimedia card (eMMC) flash memory. The IO board hosts 120 general-purpose input/output (GPIO) pins, a high-definition multimedia interface (HDMI) port, a universal serial bus (USB) port, two camera ports, and two display ports.

In order to obtain more USB ports to meet future needs, the RPi CM3+ Dev Kit is equipped with a USB hub (DUB-H4 rev E, D-Link Corporation, Taipei City, Taipei, TWN). One highlight of this USB hub is that it supports controlling USB power per port. This feature is handy for the RPi CM3+ Dev Kit to power off and restart the connected USB devices.

A 4G USB adapter (ZTE MF833V, ZTE, Shenzhen, CHN), together with a mobile broadband subscription, is used to gain access to the Internet. The supported maximum link rate is 150 Mbit/s for download and 50 Mbit/s for upload under the specific 4G network.

The Raspberry Pi Imager [38] is used to install Raspberry Pi OS for the edge device. A terminal tool called MobaXterm [39] is used to operate the RPi CM3+ via serial or secure shell session. Other terminal tools, e.g., PuTTY, can also be used as alternatives to MobaXterm. The communication between the RPi CM3+ and the microcontroller is based on the serial peripheral interface (SPI) bus. The python Linux SPI library spidev 3.5 [40] is used to implement communication functionality between the edge device and a collector. The SPI clock frequency is 122,000 Hz. The python Azure IoT Device software development kit (SDK) azure-iot-device 2.3.0 [41] is used for communicating with the Azure IoT Hub. In order to facilitate remote management and maintenance, a connection service [42] is adopted to gain access to the edge platform. A series of utilities ensure the long-term operation of the program. For instance, remote access allows researchers to log in to the edge device for management and the network monitor maximizes the network availability and prevents the device from being offline.

A plastic box (TPC 201610T/TAM 201610, Fibox AB, Bromma, Stockholm, SWE) with a dimension 163 × 201 × 98 mm is used for packaging the edge platform and the first set of a collector and five sensors. An assembled sensor box is shown in Figure 4a. In order to obtain good heat dissipation performance, a sufficient number of holes are drilled on the surface of the box to allow air circulation. To mitigate the effect of internal heat dissipation, the T&RH sensor is mounted at the farthest place from the heat source (the processor of RPi CM3+ and the AQ sensor) and is completely exposed to the external environment through a rectangular hole (see Figure 4b).

### 3.3. Cloud Platform

Microsoft Azure is one of representatives of public cloud and is used to provide services for the sensing system in this study. In addition to Microsoft Azure, other public cloud services can also be used to implement the cloud part of such a sensing system.

According to the focused tasks, the cloud platform (see Figure 5) is divided into two parts, namely the back-end and the front-end. The back-end focuses on domain logic, that is, data acquisition, analysis, and storage. The front-end focuses on providing user-friendly interfaces for users to utilize applications.

The Azure services that are used to build the back-end are as follows:“IoT Hub” [43] acts as a central message hub for reliable and secure bi-directional communication between the edge and cloud platforms. The IoT Hub supports multiple messaging patterns such as device-to-cloud telemetry, cloud-to-device messages, and invoking direct methods on devices from the cloud. In this study, the device-to-cloud telemetry is used to deliver collected environmental data from the edge platform to the cloud platform.“Event Hubs” [44] is used to build a pipeline for ingesting data in real-time. When the IoT Hub receives device-to-cloud telemetries from the edge platform, Event Hubs notifies subscribed consumers to consume the messages.“Functions” [45] is an event-driven serverless compute platform. A serverless function is implemented by utilizing Functions to consume events from Event Hubs, parse sensing data from device-to-cloud telemetry, and insert sensing data into the database.“SQL Database” [46] provides scalable storage resources and is used for storing structured data in this study. Metadata of historic buildings, metadata of edge devices, and sensing data are stored in separated tables.“Blob Storage” [47] helps to store and access unstructured data at scale. Images or documents produced in this study are stored in Blob Storage.“Web Apps” [48] facilitates deployment of web applications. The sensing system provides data visualization for collected data by using a web application deployed by Web Apps service.

For each used Azure service, the scale tier is selected by current needs (see Table A1). These services also support scale-up to meet future research needs.

In order to help facility managers and researchers view real-time and historical data, a web application is developed based on dash plotly [49]. The application mainly uses module Scattergl to visualize sensing data points as line charts. Compared to module Scatter, Scattergl has advantages such as faster speed and improved interactivity.

## 4. Case Studies

This section focuses on the practical application of the implemented sensing system in historic buildings. First, a brief description of three historic buildings chosen as case studies is provided. Then, a metric to evaluate the stability of the operating status of the sensing system is illustrated. Finally, the procedure of performing relative humidity fluctuation analysis according to a European standard is introduced.

### 4.1. Description of Three Historic Buildings

In this study, three historic buildings (see Figure 6) with different kinds of use and characteristics in Norrköping, Sweden, are chosen as case studies to test the functionalities and stability of the sensing system. These three historic buildings have been listed as protected buildings and are still open to the public for visiting and holding activities.

The City Museum (Figure 6a) is housed in old factory premises beside the Motala river, in the middle of Norrköping’s old industrial landscape. Those factory premises were erected in the 19th and 20th centuries and have been listed as protected buildings according to the national heritage legislation since 1990. The City Museum’s collection of objects includes almost 40,000 individual objects. The collections include, for example, handicraft tools of all kinds, weaving and spinning machines, billboards, printed fabrics, and sheets. These are all kinds of objects that, in different ways, illustrate Norrköping’s history, mainly the 19th and 20th century crafts and industrial history.

The City Theatre (Figure 6b) is located in the city centre of Norrköping. The building was completed in 1908 and is a representative of the Art Noveau style. The building is used as a platform for performing arts and is owned by region Östergötland and Norrköping Municipality. The building has been listed as a protected building according to the national heritage legislation since 1990.

The Auditorium (Figure 6c) was originally constructed as a church in 1827. From 1913 the building had been used as the city concert hall until 1994. Today, the Auditorium hosts local concerts and is used as a lecture hall by the local culture school. Norrköping Municipality is the owner of the building. The Auditorium was listed as a protected building according to the national heritage legislation in 1978.

Since 16 March 2021, the first batch of three sensor boxes (see Figure 7) has been deployed, one for each building. There are two key focus points for deploying the first batch of sensor boxes: One is to examine the stability of the sensing system in the actual operating environment; the other is to obtain a preliminary understanding of the variations in the indoor environment of the three historic buildings.

In the City Museum, the sensor box (Figure 7a) is deployed in an exhibition room housing ancient collections on the third floor to improve the preservation of these collections. In the City Theatre, the sensor box (Figure 7b) is deployed under the fence of the second floor of the grandstand, which is nearly the spatial center of the hall, to obtain environmental conditions that are close to the average values of the entire hall. In the Auditorium, the sensor box (Figure 7c) is deployed under the stage to measure the indoor environment while detecting human activities.

### 4.2. Metric to Evaluate the Stability

The loss rate of data samples is taken as the metric to evaluate the stability of data acquisition. The transmission of data is an end-to-end process and it involves cooperation between various modules. Therefore, the loss rate can be a good measurement of the operating stability of the entire system. The loss rate (*lr*) in a time interval is given by the following:(1)$$lr=\frac{expected-actual}{expected}\times 100\%$$
where *expected* denotes the number of data samples expected to be collected and *actual* denotes the actual number of collected data samples.

In this paper, the time interval chosen for calculating the loss rate of data samples is 56 days, from 5 April 2021 12:00 a.m. to 31 May 2021 12:00 a.m.; the timezone is Central European Time (CET). The expected number of data samples that each sensor box collects is 56×24×60×60÷15 = 322,560.

### 4.3. Fluctuation Analysis of Relative Humidity

To preliminarily analyze fluctuation of relative humidity, the method in European standard EN 15757:2010 [50] is used. This European standard serves as a guide for establishing temperature and relative humidity conditions in historic buildings to prevent climate-induced physical damage to hygroscopic and organic materials.

In this paper, a preliminary fluctuation analysis is performed for relative humidity in the City Museum because it has rich collections. First, to eliminate occasional spike noises of the relative humidity readings, a resampling was performed by taking the median of the readings every five minutes. Then, based on resampled readings, the following values are calculated according to this European standard.

Average level over a selected period: This level is calculated as the arithmetic mean of the relative humidity readings as follows:
(2)$$\bar{x}=\frac{1}{n}\sum_{i=1}^{n}x_{i}$$
where $\bar{x}$ denotes the average level, *x_i_* denotes single resampled reading, and *n* denotes the total number of resampled readings over the selected period. In this paper, the selected period is from 5 April 2021 12:00 a.m. (CET) to 31 May 2021 12:00 a.m. (CET).Monthly cycles: This cycle is determined by computing the central moving average (MA) for each reading, which is the arithmetic mean (see Equation (Equation 2) for calculation) of all the relative humidity readings recorded over a 30-day period that includes 15 days before and 15 days after the average is computed. In this paper, relative humidity readings measured from 21 March 2021 12:00 a.m. (CET) to 15 June 2021 12:00 a.m. (CET) are used to calculate monthly cycles between 5 April 2021 12:00 a.m. (CET) and 31 May 2021 12:00 a.m. (CET).Short-term fluctuations: A short-term fluctuation is defined as the difference between a current reading and the 30-day MA calculated for that reading as mentioned above. As a result, the short-term fluctuations consider both natural seasonal variability and the stress relaxation time constant of the materials.

After calculating the average level, monthly cycles, and short-term fluctuations, the target range of relative humidity can be determined by the procedure:If the relative humidity is steady, there is no need to adjust the relative humidity or temperature.If the relative humidity is unsteady, the 7th and 93rd percentiles of the fluctuations recorded during the monitoring period are used to determine the lower and upper boundaries of the target range of relative humidity, respectively.The 7th and 93rd percentiles are obtained by ordering the fluctuations from the lowest negative value to the greatest positive value and picking the values below which 7th or 93rd percent of observations are found, respectively.

In this manner, 14% of the greatest and most risky fluctuations are excluded. If the above procedure determines that the target range of relative humidity fluctuations deviates by less than 10% from the 30-day MA relative humidity level, the calculated limit is deemed unnecessary severe and the permissible short-term fluctuations can be adjusted to 10%.

## 5. Results and Discussion

This section summarizes obtained results and findings. First, the portal of the developed web application is presented. Then, the performance of system stability is shown. Finally, the fluctuation of relative humidity in the City Museum is analyzed.

### 5.1. Data Visualization and Sharing

Figure 8 shows the dashboard page of the developed web application [51] for visualizing collected sensing data. The dashboard supports viewing real-time and historical data by selecting target building (see Figure 8a) and date range (see Figure 8b).

A resampling option (see Figure 8c) is provided to improve user experience by reducing the number of data points to be rendered in a graph. Since the data are collected every 15 s, the number of data points for each environmental parameter per day is 24×60×60÷15 = 5760. Therefore, when the selected date range (see Figure 8b) is too extensive, the total number of data points to be rendered will be huge, resulting in a long page response time and, hence, affecting the user experience. When users only need to observe the overall trend of the data, the resolution of the data can be appropriately reduced. In this case, selecting a resampling option helps to display large quantities of data faster. The adopted resampling method is that, regardless of the selected date range, 720 data points are resampled uniformly according to the time distribution. For example, if one-day historical data are selected to show, then the data points will be resampled every two minutes.

Once the target building, date range, and the decision on whether to resample are selected, the dashboard page will show the historical sensing data in separated graphs (see Figure 9).

Each graph on the dashboard page is interactive. The menu bar (see Figure 10a) consists of common interactive options, such as Download plot, Zoom in/out, and Reset axes. Interaction can also be performed by directly clicking, dragging, and dropping the plot area. For instance, when moving the mouse cursor over the data point (see Figure 10b), the graph supports the revelation of more information about a data point by showing a hover label near the mouse cursor.

This study is part of a multidisciplinary cooperation project with partners from Linköping University-Campus Norrköping, Uppsala University-Campus Gotland, and the Research Institute of Sweden (RISE) in Linköping and Norrevo in Norrköping. To facilitate data sharing, a Download page (see Figure 11) is provided for sharing collected raw data between partners.

All sensed data since deployment can be downloaded from this page. The data are organized in three comma-separated values (CSV) files. The descriptions for the columns of each file are as follows:buildings.csv– Id: Primary key for buildings. Each building has a unique value;– BuildingName: Name of a building.devices.csv– Id: Primary key for edge devices. Each edge device has a unique value;– DeviceName: Name of an edge device;– BuildingId: Foreign key. References the primary key of buildings.csv.sensing.csv– Id: Primary key for sensing data. Each record has a unique value;– UtcTimestampMs: Milliseconds from 1 January 1970 at Coordinated Universal Time (UTC), for indicating when the measurement was taken;– PartitionKey: Days from 1 January 1970 at UTC, for helping partition table;– DeviceId: Foreign key, references the primary key of devices.csv;– CollectorId: Unique identification for collectors under an edge device;– Humidity: Relative humidity. The real value is dividing the raw value by 100;– Temperature: Degree Celsius. The real value is dividing the raw value by 100;– CO2: CO_2_ concentration in PPM. The raw value is the real value;– Dust: Dust concentration in pcs/L. The raw value is the real value;– AirQuality: An integer value (0–1023) mapped from output voltage (0–5 V) of the AQ sensor;– Vibration: Rising edge count in a period of 15 s.

### 5.2. System Stability

Table 1 shows that all the loss rate per sensor box between 5 April 2021 12:00 a.m. (CET) and 31 May 2021 12:00 a.m. (CET) was about 2%. Therefore, there is no significant difference in the loss rate of each sensor box. The total number of data samples expected to be collected from the three sensor boxes was 967,680, the total number of lost data samples was 19,330, and the average loss rate was 2%.

Metrics embedded in Azure IoT Hub and Functions are used to investigate when and where the loss happened. Figure 12 shows that IoT Hub received ∼964,540 messages, which means the number of the lost data samples between sensor boxes and IoT Hub was about 967,680 − 964,540 = 3140. It is speculated that the loss between sensor boxes and IoT Hub is because access to the Internet is occasionally unstable.

Figure 13 shows that the function was executed ∼947,940 times. The count of executed times is slightly less than the actual number of collected data samples (948,350). It is speculated that this difference is caused by the counting error of metrics and the count of executed times is corrected to 948,350, which means that the number of lost data samples between IoT Hub and Functions was about 964,540 − 948,350 = 16,190.

Therefore, ∼84% of data loss occurs between the IoT Hub and Functions (see Table 2). This is due to a shared service plan that was used to host the Function App and Azure does not provide service level agreement (SLA) for shared service plan [52]. The data loss can be easily mitigated by upgrading to service plans with SLA.

Overall, the sensing system can run stably for long-term monitoring of the indoor environment of historic buildings. A uniformly distributed loss rate of ∼2% is acceptable for sampling environmental data. Of course, if a lower loss rate is needed in the future, the service plan can be easily upgraded to obtain a 99.95% SLA, which can decrease the total loss rate to ∼0.3%.

### 5.3. Fluctuation Analysis of Relative Humidity in the City Museum

The measured relative humidity, calculated in a monthly cycle, and average values are shown in Figure 14. The average relative humidity between 5 April and 31 May was 25.7%. From 5 April to 10 May, most relative humidity values were lower than the average. After 10 May, the relative humidity increased and most values were above the average.

According to the obtained monthly cycle, the fluctuation of each sampling point for the monthly cycle was calculated. After sorting these fluctuations in ascending order, the obtained 7th percentile value was −4.0% while the 93rd percentile value was 4.2% (see Figure 15). The absolute values of the 7th and the 93rd percentiles are both less than 10%. Therefore, according to the standard, the allowable fluctuation range can be relaxed to 10%.

Figure 16 shows the obtained safe band and the relationship between the original measured value and the safe band. Most of the values were within the safe band. This indicates that the fluctuation of relative humidity in the City Museum is within the acceptable limit. A few points outside the safe band (around 11 May) can be regarded as abnormal points so that when detecting abnormal points, the HVAC systems can be improved to reduce fluctuations in relative humidity.

## 6. Conclusions

This paper presents a digitalization framework for smart preservation of historic buildings implemented with a sensing system for long-term monitoring of the indoor environment. By utilizing open-source software and services provided by the Azure cloud, the developed sensing system has good scalability, portability, and stability. The field testing results of the deployment of the implemented sensor boxes so far in three historic buildings in Norrköping, Sweden, have verified these advantages. The field testing results show that the implemented sensing system has a 2% end-to-end loss rate for collecting data samples and the loss rate can be decreased to 0.3%. The low loss rate indicates that the sensing system has high stability and meets the requirements for long-term monitoring of historic buildings. Due to COVID-19, these three buildings have been in a state of restricted opening to the outside world so that environmental data that are less affected by human activities have been collected. After gradual opening, further measurements and analysis, e.g., studying the impact of human activities on the indoor environment of historic buildings, will be performed. The research study will continue at least to the end of 2023. Therefore, further data analysis will be conducted and the analysis results will be used to provide facility managers and users with more applications and preventive maintenance can finally be realized.

## Figures and Tables

**Figure 1 sensors-21-05266-f001:**
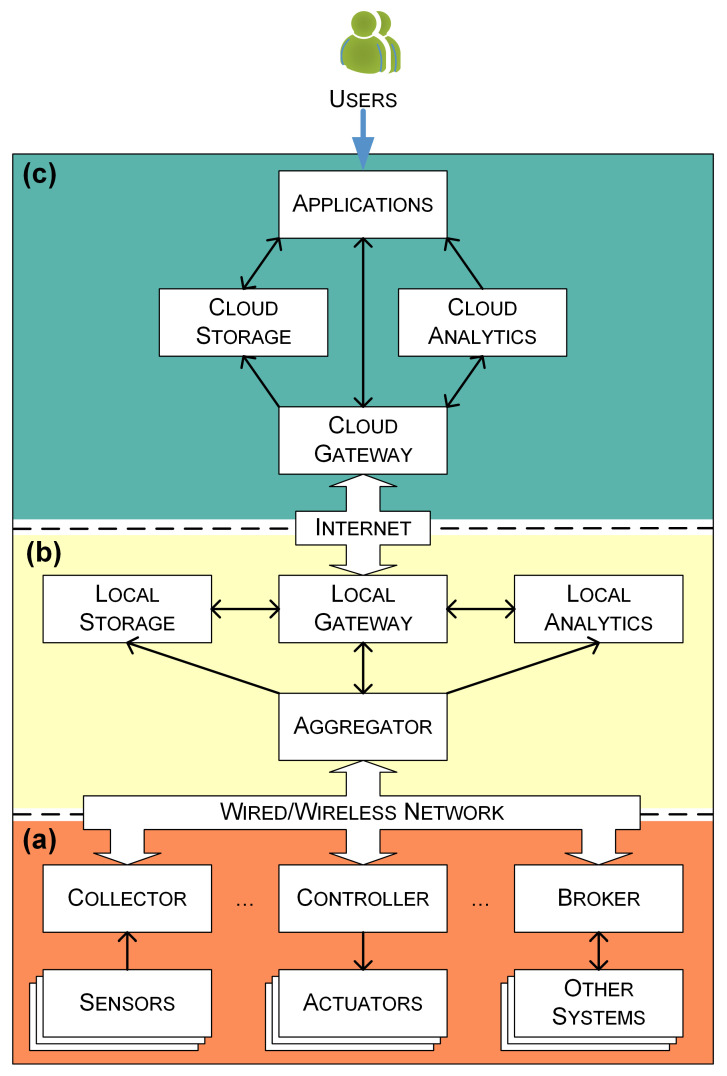
Architecture of the proposed digitalization framework. (**a**) Perception Devices, (**b**) Edge Platform, and (**c**) Cloud Platform.

**Figure 2 sensors-21-05266-f002:**
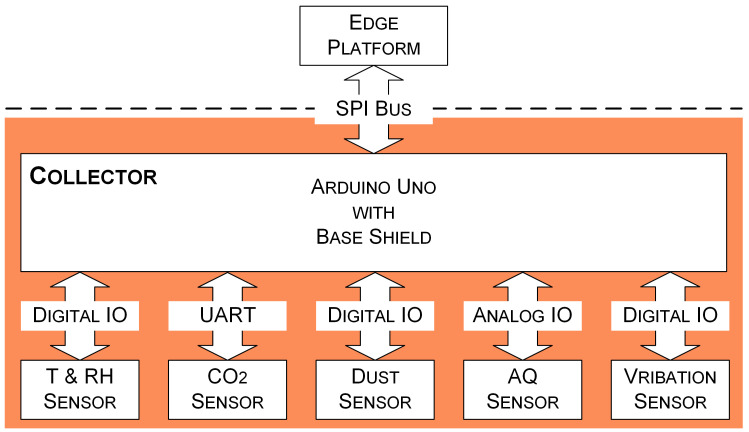
A sketch map for data communication interfaces between the collector and sensors, as well as between the collector and the edge platform.

**Figure 3 sensors-21-05266-f003:**
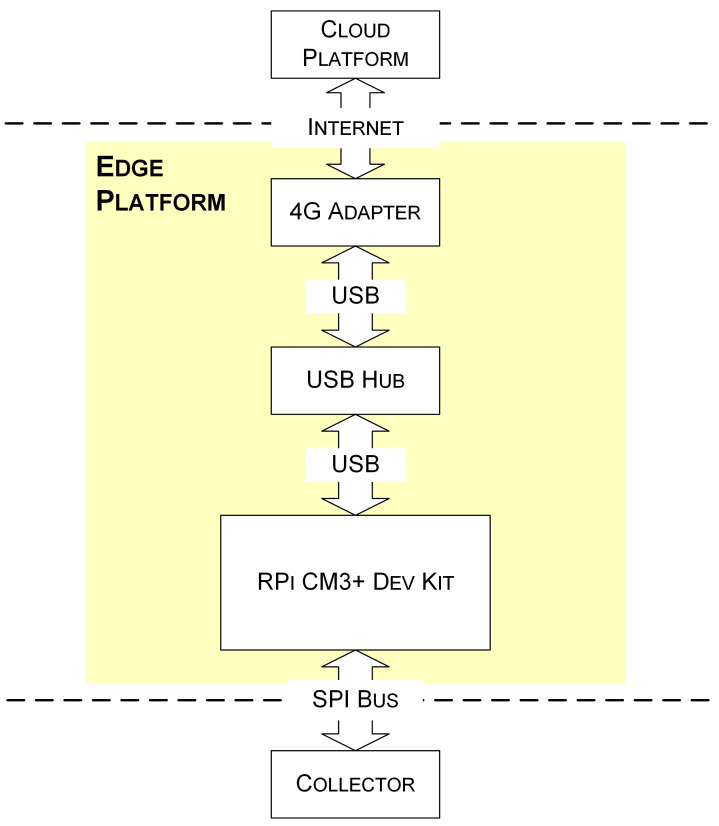
Components of the edge platform.

**Figure 4 sensors-21-05266-f004:**
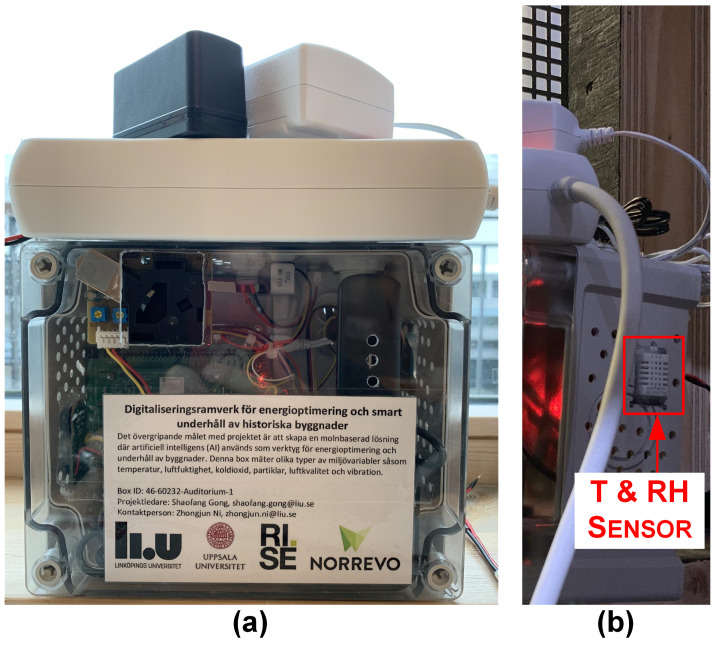
The assembled sensor box. (**a**) Front view and (**b**) right side view.

**Figure 5 sensors-21-05266-f005:**
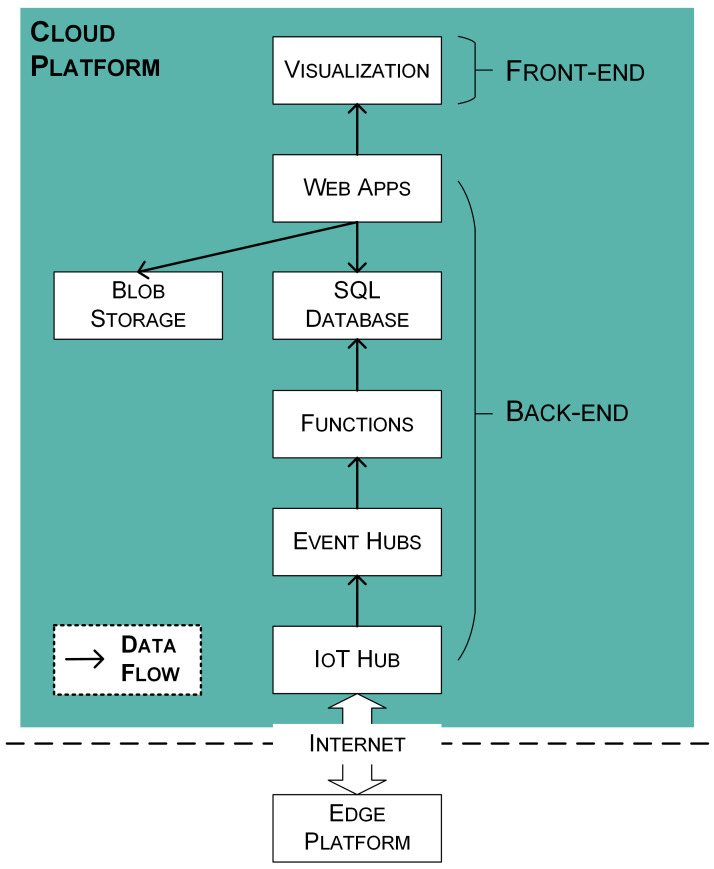
The service components used on the cloud platform and the data flow between the service components.

**Figure 6 sensors-21-05266-f006:**
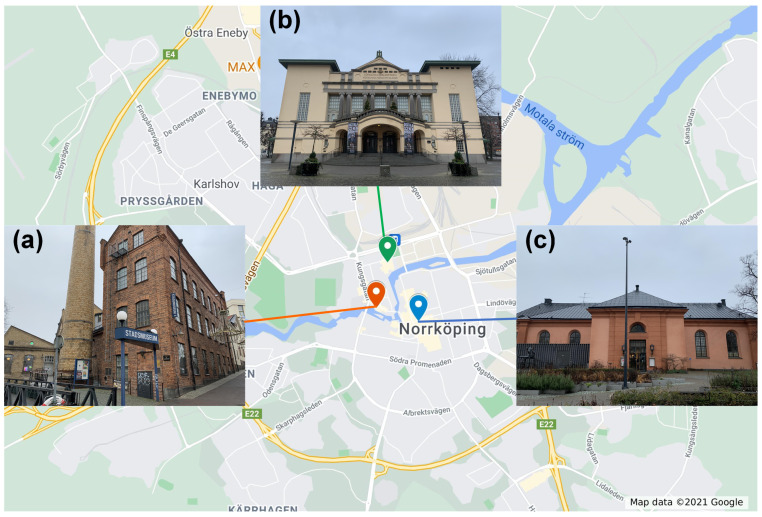
Three historic buildings in Norrköping, Sweden, are chosen as case studies: (**a**) the City Museum, (**b**) the City Theatre, and (**c**) the Auditorium.

**Figure 7 sensors-21-05266-f007:**
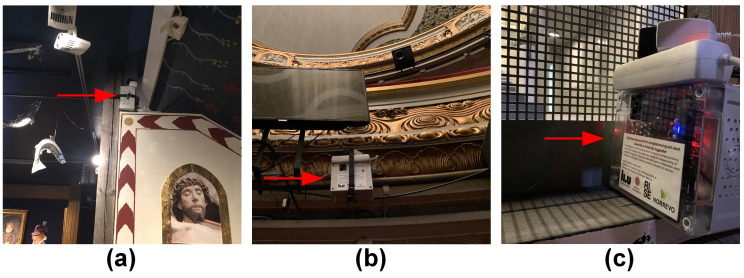
The deployments in the following: (**a**) the City Museum, (**b**) the City Theatre, and (**c**) the Auditorium.

**Figure 8 sensors-21-05266-f008:**
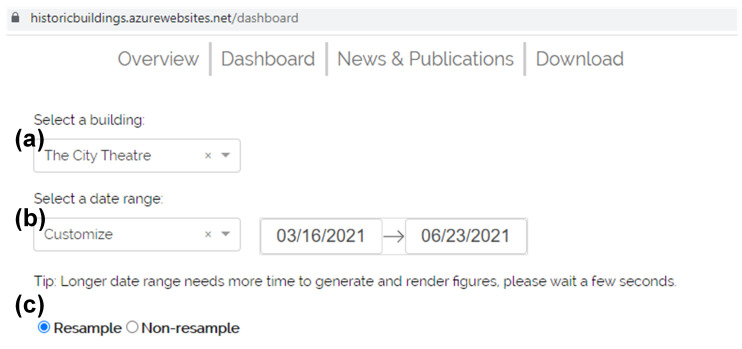
The dashboard provides data visualization for real-time and historical data by selecting the following: (**a**) the target building, (**b**) the target date range, and (**c**) whether to resample.

**Figure 9 sensors-21-05266-f009:**
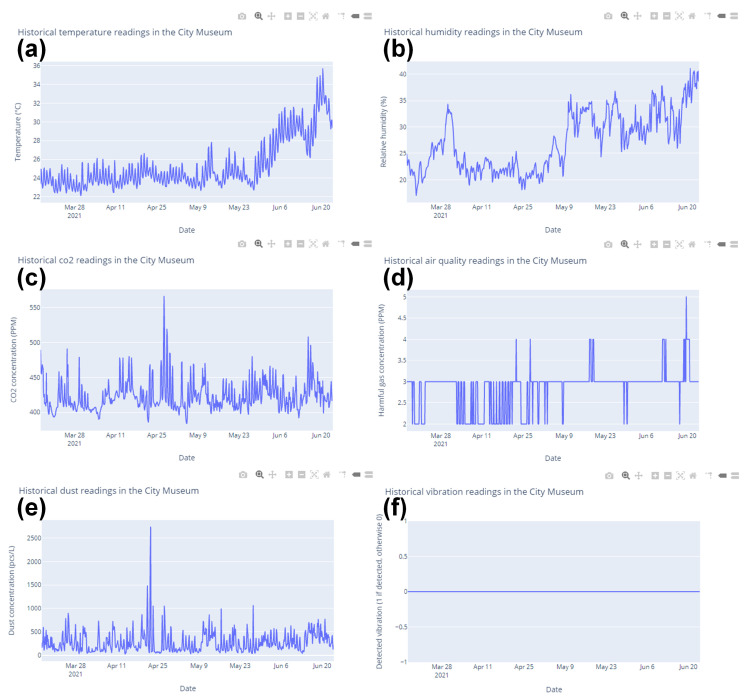
Historical sensing data collected in the City Museum from 16 March to 23 June 2021. (**a**) Temperature, (**b**) relative humidity, (**c**) CO_2_ concentration, (**d**) harmful gas concentration, (**e**) dust concentration, and (**f**) vibration.

**Figure 10 sensors-21-05266-f010:**
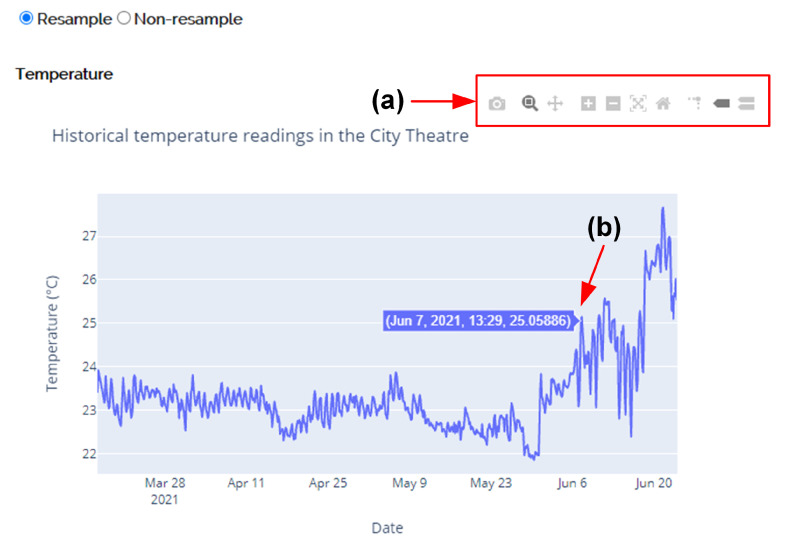
The graph supports various interactive options by the following operations: (**a**) clicking the options in the menu bar and (**b**) hovering over points on the curve.

**Figure 11 sensors-21-05266-f011:**
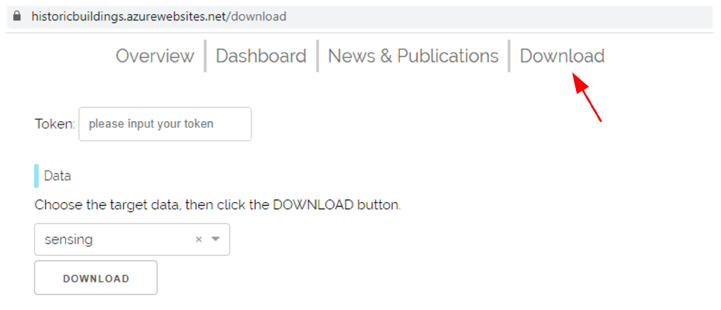
The download page supports sharing collected raw data between partners.

**Figure 12 sensors-21-05266-f012:**
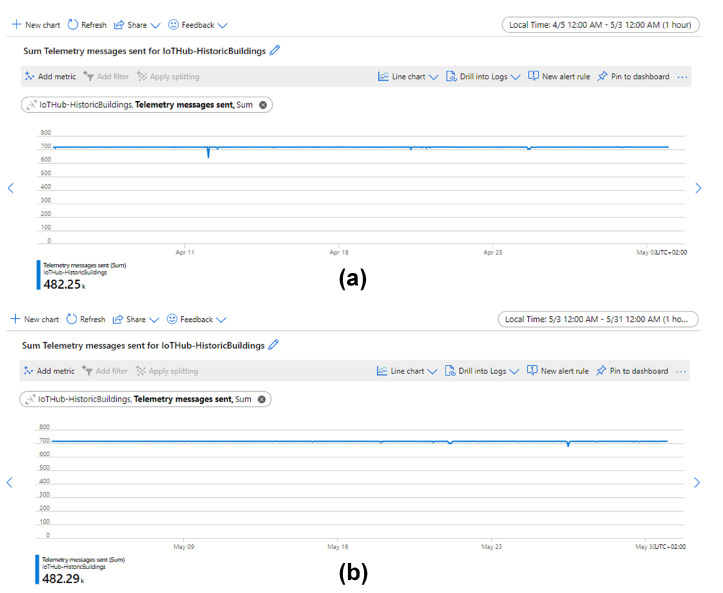
Hourly aggregated metrics for messages sent from edge to IoT Hub between 5 April and 31 May. The total number of sent messages was ∼964,540. Among them, (**a**) ∼482,250 messages were sent between 5 April and 3 May and (**b**) ∼482,290 messages were sent between 3 May and 31 May.

**Figure 13 sensors-21-05266-f013:**
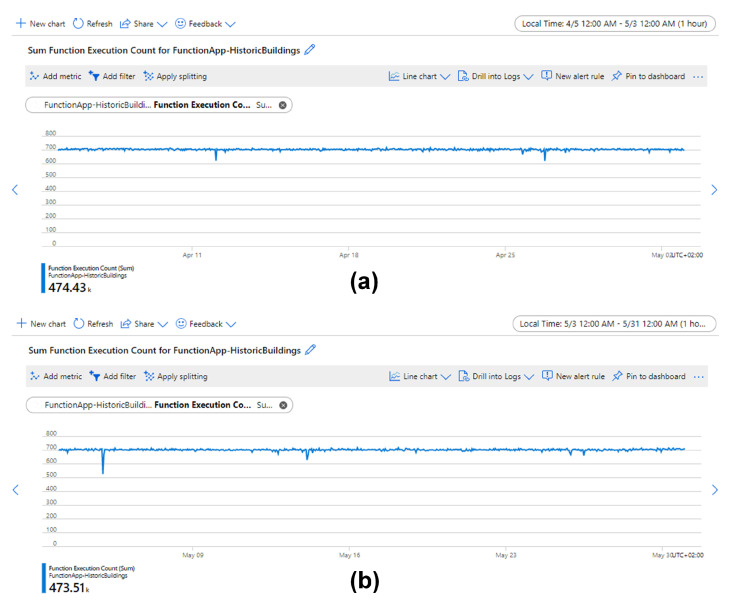
Hourly aggregated metrics for executed function count between 5 April and 31 May. The total executed function count was ∼947,940. Among them, (**a**) ∼474,430 function counts were executed between 5 April and 3 May and (**b**) ∼473,510 function counts were executed between 3 May and 31 May.

**Figure 14 sensors-21-05266-f014:**
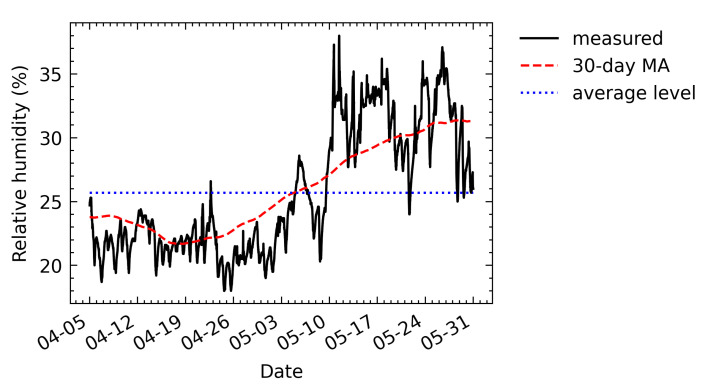
Relative humidity readings measured during eight weeks and a monthly relative humidity cycle obtained by calculating the 30-day central moving average (MA) of the readings.

**Figure 15 sensors-21-05266-f015:**
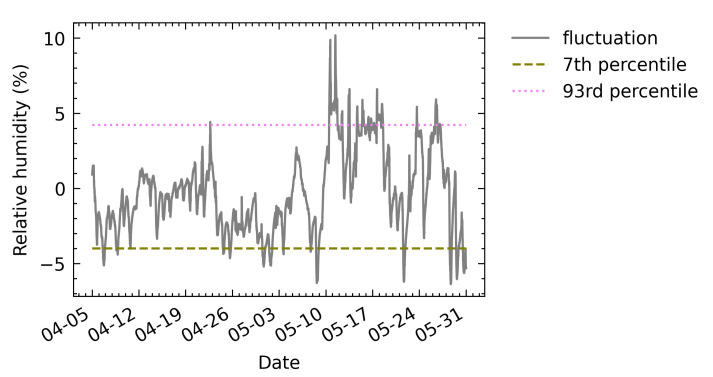
Target range of relative humidity fluctuations. Lower and upper limits of the range are calculated as the 7th and the 93rd percentiles of the fluctuation magnitudes, respectively.

**Figure 16 sensors-21-05266-f016:**
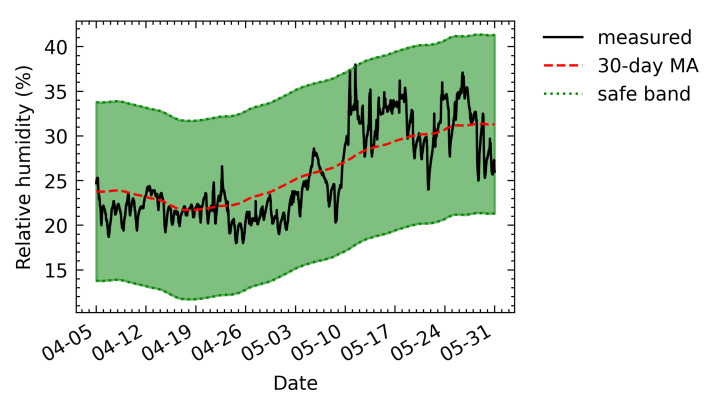
The safe band of relative humidity values for the City Museum.

**Table 1 sensors-21-05266-t001:** The loss rate of data samples per sensor box and the average loss rate of data samples for three sensor boxes.

Sensor Box Deployed In	Expected	Actual	Lost	Loss Rate
The City Museum	322,560	316,251	6309	1.96%
The City Theatre	322,560	316,121	6439	2.00%
The Auditorium	322,560	315,978	6582	2.04%
**Total**	967,680	948,350	19,330	2.00%

**Table 2 sensors-21-05266-t002:** A summary of location, amount, and proportion of lost data samples.

Location	Amount	Proportion
Edge to IoT Hub	∼3140	∼16%
IoT Hub to Functions	∼16,190	∼84%

## Data Availability

The data presented in this study are available upon request from the corresponding author. The data are not publicly available due to privacy.

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
