# Peer review of "A Sensing System Based on Public Cloud to Monitor Indoor Environment of Historic Buildings"

_sensors, 2021, doi:10.3390/s21165266_

Round 1
Reviewer 1 Report
The authors present a public cloud-based sensing system to monitor the indoor environment of historic buildings. The idea seems interesting and aims to integrate different technologies, such as IoT, Edge computing, and cloud computing. And they have deployed the proposed system in three historic buildings.
However, it is not clear (for me) the proper scientific contributions of the paper. I should say that I am not convinced on how the proposed system brings a solid contribution to the research community.
As it follows there are specific comments.
-----------------------------------------------
# Key Issues:
1. There is a problem with several figures (e.g. Fig. 1, 2, 3, and 5) in the paper and without the images is hard to understand some key elements.
2. In Section 1 (lines 47-49) the authors mention two points that could be improved comparing to related work.
The first one is the stable and scalable storage of resources. And the second one is real-time analysis.
The authors have been selected edge and cloud computing tools that could tackle these two issues. Nonetheless, no experiment has been undertaken to tackle these issues.
3. Why choose Azure? Only because has been previously used? Any references to show the efficiency of Azure would be helpful.
4. If the solution is for long-term monitoring, so some fault-tolerance test is expected, together with some other elements which would assure the solution properly works for a long-term solution.
5. What concerns at most is where is the actual new contribution of the paper?
From lines 90 to 101 the intended contributions are enumerated.
# 1. the long-term monitoring
>> This one I have already mentioned above.
# 2. implemented a sensor box
>> Ok, the paper presents the deployment of the sensor box. But, for me, this is a simple implementation of a sort of "toolbox".
>> Besides, as far as I understood, not all kinds of sensors have been used. The proposed architecture describes five different types of sensors, but only two seem to be used.
# 3. cloud solution
>> Ok, there is the implementation of a cloud solution, but again does not seem to be something new. The authors have already used an Azure solution in similar projects.
# 4. apply the system in the historic buildings
>> This seems to be the only new contribution, which is the application of the system in the three buildings.
>> But, honestly this does bring many practical results. In the case studies (previous seen) is only presented how some results on the loss rate of the sensors and the relative humidity have been obtained.
In summary, I am not convinced that to obtain these data concerning relative humidity in the buildings, the proposed architecture with edge and cloud computing is necessary. It seems too much effort for a task that is not that complex. At least, the obtained results described in Section 4 show that.
-----------------------------------------------
As it follows, some additional comments and corrections:
# 1. Introduction
> Needs to mention edge computing in the Introduction .. has it been used? Line 94 only mentions "edge device".
# 2. System Design and Implementation
> This section brings a lot of technical issues, it explains like a "toolkit" you want to deploy. But, how about the expected scientific contribution of a proposed architecture? I think the text does not present such important elements.
> I believe Section 2 seems the kind of content expected for Section 3, where the case studies are described.
# 5. Conclusions
> The authors could discuss what could be improved in the work for future developments. Could the solution presented in this paper be extended and applied to other kinds of buildings, not necessarily historic buildings?
Reviewer 2 Report
- Interesting research in a decent manuscript that needs substantial revisions. A particular strength is the application on Norrköping buildings.
- Abstract is okay but is not likely to entice the readership to continue reading the rest of the manuscript.
- Results are only presented in weak, qualitative fashion. Highest quality expression of main conclusions or interpretations is quantitative results discussed in the broadest context possible, e.g., percent performance improvement compared to a declared benchmark. “…proposed sensing system has high stability and meets the requirement…” is very weakly stated results compared to “…xxx percent performance improvement was achieved….” which seems to be immediately available from the results presented in section 4.
- Introduction is decently done with elaborate citation especially of state-of-the art methods, yet some omitted very recent literature particularly referencing competing approaches that should be presented as options for future research for the reader. Particularly in agreement with the authors assertions that real-time analysis is superior to post-facto analysis, many alternatives exist for learning algorithms, and some of them are deterministically optimal. These alternatives should be included in the literature review to give the reader an understanding of the proposed approach in the context of competing alternatives, and the reviewer has provided two recommendations below.
- Real-time optimization has recently been implemented as virtual sensoring formulated as a Hamiltonian system, while real-time application of optimal feedforward approaches are the provenance of flexible space robotic control. Both illustrations of efficacy stem from Soviet mathematician Lev Pontryagin’s minimization of Hamiltonian systems to derive sensors and controls that can be made adaptive or learning. The method is an alternative to the proposed instantiation of the sensor box of environmental sensors, collector, and edge device.
- Deterministic artificial intelligence has recently been applied to vehicles, motors, global temperatures, and even electrical vehicle sales, where the method was proposed in 2020 for unmanned vehicles but expressed as a systems approach by Shah in 2021. The approach is a competing alternative to using cloud solution for real-time monitoring and data visualization.
- Equations are substantially absent except for stability metric evaluation by the loss rate, which is scientifically sound and well presented, enhancing the manuscript quality.
- Figures are decently done with some mandatory improvements to ensure the readership has access to the content.
- Figures 1,2,3,5,6,7,8,9,10,11,12 (all but figure 4) are missing, while their captions are presented.
- Figures 13-15 are very well presented.
- Tables are decently done, but scarce and substantial quantitative results are neglected (not presented in table for emphasis and ease of reading).
- Inclusion of a table defining variables and acronyms in an appendix is welcome and effective. Please add such including at least IoT Hub, Functions, SQL Database, Web Apps, Collector and Sensors, Edge Platform, Cloud Platform, PWM, UART, IO, D2-D8, I2C, PM, CO2, T&RH, DHT22, RH, ……..etc.
- The individual components of the proposed system are decently described aiding repeatability.
Round 2
Reviewer 1 Report
The authors have provided proper answers for all addressed issues in the review. The contributions are mainly presented, and the missing figures were added to the paper; consequently, several issues can be properly understood.
Reviewer 2 Report
Thank you for improving the manuscript. Figures 9, 10, 12, and 13 remain illegible due to interior font size. As a general guide to aid the authors, the reviewer recommends noting the smallest font size permissible in the manuscript template is the figure caption (providing a proximal example to guide sizing figures to insure legibility).